# Diagnosis of *Helicobacter pylori* Infection and Recent Advances

**DOI:** 10.3390/diagnostics11081305

**Published:** 2021-07-21

**Authors:** Hang Yang, Bing Hu

**Affiliations:** Department of Gastroenterology, West China Hospital, Sichuan University, Chengdu 610041, China; 15841123892@163.com

**Keywords:** diagnosis of *Helicobacter pylori*, endoscopy, artificial intelligence, polymerase chain reaction

## Abstract

Background: *Helicobacter pylori* (*H. pylori*) infects approximately 50% of the world population. Its infection is associated with gastropathies, extra-gastric digestive diseases, and diseases of other systems. There is a canonical process from acute-on-chronic inflammation, chronic atrophic gastritis (CAG), intestinal metaplasia (IM), dysplasia, and intraepithelial neoplasia, eventually to gastric cancer (GC). *H. pylori* eradication abolishes the inflammatory response and early treatment prevents the progression to preneoplastic lesions. Methods: the test-and-treat strategy, endoscopy-based strategy, and screen-and-treat strategy are recommended to prevent GC based on risk stratification, prevalence, and patients’ clinical manifestations and conditions. Challenges contain false-negative results, increasing antibiotic resistance, decreasing eradication rate, and poor retesting rate. Present diagnosis methods are mainly based on invasive endoscopy and noninvasive laboratory testing. Results: to improve the accuracy and effectiveness and reduce the missed diagnosis, some advances were achieved including newer imaging techniques (such as image-enhanced endoscopy (IEE), artificial intelligence (AI) technology, and quantitative real-time polymerase chain reaction (qPCR) and digital PCR (dPCR). Conclusion: in the article, we summarized the diagnosis methods of *H. pylori* infection and recent advances, further finding out the opportunities in challenges.

## 1. Introduction

*Helicobacter pylori* (*H. pylori*) infection is chronic and usually acquired in childhood. Globally, *H. pylori* infects an estimated 50% of the global population which is influenced by socioeconomic status, sanitation, regions, and age. For continents, it was reported that Africa had the highest prevalence of *H. pylori* infection (70.1%), whereas Oceania had the lowest prevalence (24.4%). For countries, the prevalence of *H. pylori* infection varied from as low as 18.9% in Switzerland to 87.7% in Nigeria [1]. One meta-analysis reported an overall prevalence of 44.3% involving 410,879 participants from 73 countries in six continents, with a rate of 50.8% in developing countries compared with 34.7% in developed countries, 42.7% in females compared to 46.3% in males, and 48.6% in adults (≥18 years) compared to 32.6% in children [2]. *H. pylori* gastritis was defined as an infectious disease and should be offered eradication therapy. If there is *H. pylori*-associated dyspepsia or functional dyspepsia, eradication of *H. pylori* is the first-line treatment. Symptoms can be attributed to *H. pylori* gastritis if sustained symptoms get remission after 6–12 months [3,4]. Regarding gastric cancer (GC), some potential changes caused by *H. pylori* infection may contribute to the progress of GC, which includes gastric dysbacteriosis [5], changing gastric mucosal, and cellular immunity as one component of inflammatory microenvironment [6,7], aberrant deoxyribonucleic acid (DNA) methylation [8], abnormal expression of ribonucleic acids (RNAs) (micro RNAs, long noncoding RNA, and messenger RNAs) [9,10], and single-nucleotide polymorphisms [11], et al. Chronic atrophic gastritis (CAG) and intestinal metaplasia (IM) are precancerous conditions in which dysplasia (neoplastic precancerous lesion) and adenocarcinoma may occur. GC incidence of mild, moderate, and severe atrophy is 0.04–0.10%/year, 0.12–0.34%/year, and 0.31–1.60%/year, respectively [12,13]. GC incidence in patient with IM is 0.038–1.708%/year, and the progressing rate to dysplasia in IM patient was estimated to be 1.251%/year [14,15]. Endoscopic assessment, *H. pylori* infection diagnosis, and surveillance are recommended in patients with precancerous conditions. Endoscopically visible lesion harboring low- or high-grade dysplasia or GC should undergo staging and treatment [16,17]. *H. pylori* eradication heals acute inflammation and nonatrophic chronic gastritis and may lead to regression of atrophic gastritis and reduce the risk of GC in patients with nonatrophic and atrophic gastritis. *H. pylori* eradication is recommended in patients who have family history of GC, CAG, IM, dysplasia, or cancer and in patients with gastric neoplasia or early GC after endoscopic therapy or by subtotal gastrectomy to prevent metachronous recurrence [16,17]. Diagnosis is one of the cores of *H. pylori* management and the prevention of GC. On the aspect of potential changes associated with *H. pylori* infection mentioned above, these changes are “invisible” and need more further research to achieve the translation of their visualization from basic study to clinical practice, similar with present diagnosis methods. In this review, we concluded the diagnosis methods of *H. pylori* infection and recent advances including endoscopic diagnosis and laboratory diagnosis in detail and hope to improve above issues from the aspect of diagnosis methods.

## 2. Challenges of *H. pylori* Management and Recommended Detecting Strategy

The maximum benefit of *H. pylori* eradication is obtained if it is done while the mucosal damage is still nonatrophic [3]. The most common regimens as first-line treatment of *H. pylori* are the clarithromycin-containing triple therapy extended for more than 7 days and the nonbismuth (sequential and concomitant) and bismuth quadruple therapies [18]. The traditional standard triple therapy is associated with antibiotic resistance, which can further undermine its efficacy and result in low eradication rate [18]. Bismuth-containing quadruple therapy is confirmed as an effective regimen for eradicating *H. pylori*, especially in strains with antibiotic resistance [19,20]. The choice of *H. pylori* eradication regimen should be based on the local prevalence of clarithromycin resistance and the previous use of macrolides [19]. Decreasing eradication rate because of antibiotic resistance emerged as a main clinical problem. One meta-analysis including 178 studies from 65 countries in World Health Organization Regions reported primary and secondary resistance rates to clarithromycin, metronidazole, and levofloxacin were ≥15% [21]. A prospective study investigated in 24 centers from 18 European countries reported that primary antibiotic resistance of *H. pylori* was 21.4% for clarithromycin, 15.8% for levofloxacin, and 38.9% for metronidazole, associated with the consumption in the community of macrolides and intermediate-acting macrolides [22]. One research covering 176 articles from 24 countries in the Asia-Pacific region showed primary *H. pylori* resistance rates were 17% for clarithromycin, 44% for metronidazole, 18% for levofloxacin, 3% for amoxicillin, and 4% for tetracycline [23]. In China, primary resistance rate to clarithromycin was 20–50% [24]. The resistance rate of clarithromycin in Korea was 17.8–31.0% [25], and the clarithromycin resistance rate in Japan was 38.5% [26]. Resistance to clarithromycin was significantly associated with failure of clarithromycin-containing regimens. It should be avoided in countries where clarithromycin resistance (>15%) or proven high local eradication rates (<80–85%) [27]. Local surveillance networks are required to select appropriate eradication regimens for each region. Before eradication, test clarithromycin resistance in advance is acceptable. For example, in the Maastricht V/Florence Consensus Report, clarithromycin susceptibility testing is recommended to be performed either via a standard method (antibiogram) after culture or by a molecular test directly on the gastric biopsy specimen, when a standard clarithromycin-based treatment is considered as the first-line therapy [28]. PCR or sequencing is newly recommended to test resistance [25]. However, cost, accuracy and availability are also factors influencing resistance test and further evaluation is required. After eradication, it is recommended that all patients should be reassessed to confirm eradication, and recommended tests include the urea breath test (UBT) and the monoclonal *H. pylori* stool antigen test (Hp SAT) [4,28]. However, retesting rate ranged from 30% to 70% [29]. Poor retesting rate is also a concern in clinical practice, maybe attributed to patients’ medical compliance, ages, test convenience, potential risk of indication diseases for *H. pylori,* or followup system, etc. For example, patients receive eradication because of a family history of GC, precancerous conditions or lesion, early GC, gastric mucosa-associated lymphoid tissue (MALT) lymphoma or complicated peptic ulcer diseases (bleeding, perforation, obstruction), they will be more willing to retest the primary eradication effect. In addition, patients who failed primary eradication also had a significantly higher risk of future hospitalization for nonvariceal upper gastrointestinal bleeding, particularly among older patients and selective serotonin reuptake inhibitors users [30,31]. *H. pylori* recurrence remains another problem. A meta-analysis including 132 studies from 45 countries reported the global annual recurrence, reinfection and recrudescence rate of *H. pylori* were 4.3%, 3.1% and 2.2%, respectively, and they were associated with socioeconomic and sanitary conditions [32]. Aimed at different people for managing *H. pylori*, there are three strategies. A test-and-treat strategy is appropriate for uninvestigated dyspepsia with noninvasive tests such as UBT, which is preferred rather than prescribing proton pump inhibitor (PPI) or endoscopy [33,34,35]. An endoscopy-based strategy followed by biopsies of Sydney system should be considered in patients with dyspeptic symptoms, patients with alarm symptoms, or older patients, particularly in low prevalence *H. pylori* populations (<10%). A screen-and-treat strategy is recommended in communities at high risk of GC [36].

## 3. Endoscopic Diagnosis

### 3.1. Conventional White Light Imaging (WLI)

Globally, the prevalence of gastritis is near 50%, which was shown from 40.7% to 56.0% and included 20–30% chronic atrophic gastritis. *H. pylori*-negative gastritis was from 17.7% to 20.5%, in which chronic gastritis accounted for 10–15% [37,38,39]. It indicates that *H. pylori* infection is generally consistent with the prevalence of gastritis and *H. pylori*-positive gastritis generally accounts for more than 80%. Therefore, it is the basis of clinical application of gastritis in Kyoto classification, as only a small proportion of gastritis may not be infected by *H. pylori*. Endoscopic findings of conventional white light imaging (WLI) can initially predict the status of *H. pylori* and the suspicious infection according to gastritis in Kyoto classification, and then biopsies are taken according to Sydney system [3,40]. Kyoto classification of gastritis including diffuse redness, regular arrangement of collecting venules (RAC), fundic gland polyp (FGP), atrophy, xanthoma, hyperplastic polyp, map-like redness, intestinal metaplasia, nodularity, mucosal swelling, white and flat elevated lesion, sticky mucus, depressive erosion, raised erosion, red streak, and enlarged folds. Regarding validation research, RAC, FGP, and red streak were demonstrated with satisfactory diagnostic odds ratios (DOR) for predicting uninfected status. Nodularity, diffuse redness, mucosal swelling, enlarged fold and sticky mucus were significantly associated with current infection. Map-like redness was responsible for past infection, and the overall diagnostic accuracy rate of Kyoto classification of gastritis was more than 80% [41,42,43,44]. Furthermore, with regard of uninfected status, one study showed RAC had excellent negative predictive value (NPV) of about 90% and sensitivity value of up to 85% [45]. A meta-analysis including 4070 patients also showed RAC was a valuable endoscopic feature of uninfected status with 0.80 sensitivity, 0.97 specificity, and 0.97 area under the curve (AUC) [46]. With regard of current infection, Kyoto classification score (including atrophy, IM, enlarged folds, nodularity, and diffuse redness) ≥2 could predict *H. pylori* infection with 89.7% accuracy, 78.3% sensitivity, and 92.0% specificity in patients with a high-negative titer of anti-*H. pylori* antibody [47]. One study showed an AUC for *H. pylori* infection of WLI was 0.81 in the corpus and 0.71 in the antrum and indigo carmine contrast (IC) method was useful in gastric swelling areas [48]. Other research reported 0.82–0.92 AUC used self-assembled score systems to predict *H. pylori* infection [49,50]. However, there are two problems that cannot be ignored in real time clinical practice. The first one is the professional level and experience, as well as interobserver agreement. A brief mini-lecture on the Kyoto Classification of Gastritis could improve the accuracy from 90.3% to 96.5% [51]. The second one is the clinical routine that biopsy rather than other detecting methods (UBT, Hp SAT, or serological test) will be taken after primary prediction via Kyoto Classification of Gastritis. From the data mentioned above, Kyoto Classification of Gastritis is more characterized with higher specificity and slightly inferior sensitivity. One clinical research reported no endoscopic features (alone or in combination) showed a sensitivity of more than 57% for *H. pylori* infection [52], which may further result in increasing missed diagnosis rate. The uneven distribution of *H. pylori* inevitably leads to sampling errors in biopsy-based examinations including rapid urease test (RUT), histology, or culture. Biopsies from multipoints can improve the accuracy of detection. Two samples (one from the antrum avoiding areas of ulceration and obvious IM and one from normal appearing corpus) can provide the highest yield for RUT, as well as time saving [53]. The sensitivity of RUT was reported to vary between 80% and 100%, and its specificity is between 97% and 99% [54]. If less than 10^4^ bacterial cells are present in the gastric biopsy, false-negative results are obtained most probably [55]. It is essential to improve the sensitivity. Therefore, many efforts were done on newer imaging techniques such as image-enhanced endoscopy (IEE) and aiding systems such as AI.

### 3.2. Image-Enhanced Endoscopy (IEE)

IEE including magnifying endoscopy and digital chromoendoscopy such as narrow-band imaging (NBI), autofluorescence imaging (AFI), blue laser imaging (BLI), and linked color imaging (LCI) offered advantages in diagnosing *H. pylori*.

Magnifying endoscopy (ME) can provide more precise information concerning the collecting venules, the network of capillaries surrounding the gastric pits, the swelling of the surface epithelium between pits, and the enlargement and destruction of the pits, which was considered useful for the diagnosis of histopathologic gastritis [56,57]. Type Z-0: subepithelial capillary network (SECN) with regular arrangement of collecting venules and gastric pits resembling pinholes. The sensitivity, specificity, positive predictive value (PPV), and NPV of the type Z-0 pattern for predicting normal gastric mucosa were 90.3–92.7%, 93.9–100%, 100%, and 83.8% [58,59,60]. Types Z-1 and 2 patterns (enlarged gastric pits, irregular or loss of SECN, and an absence of collecting venules) were reported with sensitivity, specificity, PPV, and NPV for predicting *H. pylori* infection were 100%, 92.7%, 83.8%, and 100% [58,59]. A meta-analysis involving 1897 patients reported the pooled sensitivity and specificity of ME to predict *H. pylori* infection were 0.89 and 0.82, respectively, with an AUC of 0.95 [61]. Compared with that of conventional WLI, ME can be superior for the diagnosis of *H. pylori* gastritis. The “pit plus vascular pattern” classification in the gastric corpus observed by ME was able to accurately predict the status of *H. pylori* infection with a pooled sensitivity and specificity of 0.96 and 0.91, respectively, with an AUC of 0.99 [61]. The sensitivity and specificity of irregularly arranged antral ridge pattern for the prediction of antral gastritis were 89.3–96.3% and 65.2–73.7%, respectively [60,62]. Indigo carmine staining increased sensitivity and specificity up to 97.6% and 100% for corporal gastritis, and up to 88.4% and 75.0% for antral gastritis, respectively [60].

### 3.3. Electronic Chromoendoscopy

Non-M-NBI endoscopy is an optical image enhancement technique to enhance the visualization of mucosal microscopic structure and capillaries of the superficial mucosal layer. One study firstly and retrospectively found NBI could be a promising method for *H. pylori* infection identification [63]. According to five gastric mucosal morphologic patterns of non-M-NBI, type 3 (rod-shaped gastric pits with prominent sulci), 4 (ground glass-like morphology), or 5 (dark brown patches with bluish margin and irregular border) morphologies were statistically significant in predicting *H. pylori* positive status and achieved 94.28% sensitivity, 96.66% specificity, 98.50% PPV, and 87.87% NPV [64]. A further retrospective study on the site-specific biopsy guided by NBI of abnormal mucosa rather than the random biopsy for the diagnosis of *H. pylori* showed higher 95.4% sensitivity and 97.3% specificity [65]. However, a multicenter prospective study demonstrated no difference in the accuracy of diagnosing *H. pylori* gastritis between NBI and WLI (74% NBI vs. 73% WLI), although NBI demonstrated slightly higher sensitivity (69% vs. 57) but reduced specificity (67% vs. 79%) [66].

M-NBI endoscopy clearly visualizes superficial gastric mucosal patterns and capillary patterns. In one study including 106 patients, gastric corpus mucosal patterns observed by M-NBI were divided into the following categories: normal: small, round pits with regular subepithelial capillary networks; type 1: slightly enlarged, round pits with unclear or irregular subepithelial capillary networks; type 2: obviously enlarged, oval or prolonged pits with increased density of irregular vessels, and type 3: well-demarcated oval or tubule-villous pits with clearly visible coiled or wavy vessels. *H. pylori* infection positive ratios of normal and types 1, 2, and 3 patterns were 7.5%, 92.9%, 94.5%, and 66.7%, respectively [67]. In another study including 90 patients, the mucosa of the gastric antrum was observed by M-NBI, and the gastric microstructure was categorized into five types (A–E) (type B: elongated open branch-like pits with regular microvasculature; type C: dilated pits and increased branching microvasculature). The sensitivity and specificity of type B alone, type C alone, and types B + C for the detection of *H. pylori* infection were 52.2% and 87.%, 22.8% and 92.2 %, and 75.0% and 79.1%, respectively [68]. Compared with that of WLI, the sensitivity, specificity, PPV, and NPV of M-NBI were higher (0.91 vs. 0.79, 0.83 vs. 0.52, 0.88 vs. 0.70, and 0.86 vs. 0.63, respectively) in a study with 56 patients after ESD [69].

### 3.4. Linked-Color Imaging and Blue Laser Imaging

Linked color imaging (LCI) can show mucosal color similar to WLI but produce more color patterns of the mucosa due to emission intensity at wavelengths different from WLI [70]. These colors allow endoscopists to diagnose a variety of lesions such as inflammation areas because of the high color contrast with surrounding mucosa. Blue laser imaging (BLI) is another IEE that combines narrow-spectrum blue laser with white light to make up the deficiency of NBI [71]. The push of a single button during endoscopy allows one to switch between LCI and BLI. LCI is brighter than WLI, and BLI is brighter than NBI. LCI produces particularly bright images in the stomach and is useful when screening gastric lesions, whereas BLI-bright and BLI are also useful in displaying mucosal structure and vessels in close-up views inside the stomach, as well as relatively close views, especially the antrum [72]. Some research has indicated *H. pylori* infection could be identified by LCI and BLI. With regard of BLI, one study included patients’ mucosal patterns observed by BLI and divided into Spotty, Cracked, and Mottled pattern groups with results of 12/77, 105/17, and 138/90 negative/positive for *H. pylori* infection, respectively. The specificity and PPV for endoscopic diagnosis with positive *H. pylori* infection based on the Spotty pattern were 95.3% and 86.5% [73]. On the aspect of LCI which is more suitable in wide-lumen organ than BLI, studies based on Kyoto Classification of Gastritis to assess the visibility of LCI, WLI, and BLI found that LCI could improve visibility especially for diffuse redness, spotty redness, map-like redness, patchy redness and red streaks [74,75,76]. When compared with that of WLI, LCI could identify *H. pylori* infection by enhancing endoscopic images of the diffuse redness of the fundic gland and achieve more optimal diagnostic power (accuracy 85.8% vs. 74.2%, sensitivity 93.3% vs. 81.7%, and specificity 78.3% vs. 66.7%) [77]. Another study reported that the application of LCI at the corpus to identify *H. pylori* infection could be reliable and superior to WLI with the highest accuracy among groups (81.2% vs. 64.3–76.5%), as well as higher sensitivity (85.41%) and specificity (79.71%) [78]. A prospective study also indicted the accuracy of LCI was higher than that of WLI (accuracy 86.6% vs. 79.5%, sensitivity 84.4% vs. 84.4%, and specificity 88.9% vs. 74.6%) [79]. When compared with ME, one study recruiting 122 patients (36 had *H. pylori* infection) showed that LCI could play a similar role with ME and demonstrated diagnostic abilities of *H. pylori* infections by LCI (78.38% accuracy, 70.97% sensitivity, 82.5% specificity, 59.46% PPV and 87.84% NPV), ME (81.98% accuracy, 81.25% sensitivity, 83.87% specificity, 64.10% PPV and 91.67% NPV), and both LCI and ME (78.38% accuracy, 80.65% sensitivity, 76.25% specificity, 57.78% PPV, and 92.42% NPV) [80].

i-Scan digital chromoendoscopy is also a digital contrast method to enhance minute mucosal structures and subtle changes in color [81]. One prospective study showed the overall diagnostic accuracy of i-scan was higher at 97% compared to 78% of WLI, and a greater proportion of patients were identified as endoscopic features of *H. pylori* under i-scan examination (79/146 vs. 45/146) [82]. Another research reported that the type 2 + 3 patterns of M-i-scan was superior to ME for the prediction of *H. pylori* infection in 84 patients (accuracy: 94.0% vs. 84.5% and specificity: 93.5% vs. 80.6%), while the sensitivity of the two modes was the same (95.5%) (type 2 meant honeycomb type SECN with regular round pits with or without sulci, in the absence of collecting venules, and type 3 meant loss of normal SECN and collecting venules, with white enlarged pits surrounded by erythema) [83].

Confocal laser endomicroscopy (CLE) is a new endoscopy technique for subsurface analysis of the gastric mucosa and in vivo histology examination during endoscopy. CLE was used for the first time to detect *H. pylori* in vivo reported in 2005 [84]. In a prospective study, CLE image criteria for *H. pylori* infection were established in a pilot study of 20 patients, and images of 83 consecutive patients was observed by CLE with any of the three features (white spots, neutrophils and micro-abscesses) with 92.8% accuracy, 89.2% sensitivity, and 95.7% specificity for predicting *H. pylori* infection in vivo during endoscopy [85].

### 3.5. AI: One of Present Advances in Endoscopic Diagnosis of H. Pylori Infection

In the field of endoscopy, the application of AI has received wide attention including gastrointestinal cancers and benign diseases based on endoscopic images, videos and histopathologic slides [86]. *H. pylori* infection, as a dominant cause of CAG and GC, was also detected via AI methods based on endoscopic images. One meta-analysis including 8 studies and 1719 patients (385 patients with *H. pylori* infection vs. 1334 controls) diagnosed by WLI, BLI, or LCI reported that the sensitivity, specificity, DOR, and AUC of AI for the prediction of *H. pylori* infection were 0.87, 0.86, 40, and 0.92, respectively. The accuracy of the AI algorithm reached 82% for discrimination between noninfected images and posteradication images [87]. Regarding WLI, a DCNN model trained and verified by WLI of gastric antrum showed a power in diagnosing atrophic gastritis with 94% accuracy, 0.95 sensitivity, and 0.94 specificity, which were higher than those of experts [88], and AI diagnosis could be done in a considerably shorter time less than 200 s [89,90]. On the aspect of ME, a CNN system was pretrained using 1492 early gastric cancer (EGC) and 1078 *H. pylori* associated gastritis images from M-NBI to differentiate between EGC and gastritis and evaluated by a separate test data set (151 EGC and 107 gastritis images based on ME-NBI). Finally, it achieved a diagnostic ability with 85.3% accuracy, 95.4% sensitivity, 71.0% specificity, 82.3% PPV and 91.7% NPV, respectively, and 51.83 images/second overall test speed (0.02 s/image) [91]. In terms of LCI, a study developed a machine learning method to diagnose *H. pylori* infection with 87.6% accuracy, 90.4% sensitivity, 85.7% specificity, 80.9% PPV and 93.1% NPV [92]. One study developed two different CAD systems, one for LCI (LCI-CAD) and one for WLI (WLI-CAD) and achieved a comparable diagnostic accuracy to that of experienced endoscopists and a higher diagnostic accuracy of the LCI-CAD system (84.2% for uninfected, 82.5% for currently infected, and 79.2% for posteradication status) than that of WLI-CAD [93]. Another study used GoogLeNet, a 22-layer DCNN pretrained by BLI-bright and LCI and tested by 222 patients (105 *H. pylori*-positive) to achieve a significantly higher diagnostic ability of *H. pylori* infection from BLI-bright (0.96 AUC, 96.7% sensitivity, and 86.7% specificity) and LCI (0.95 AUC, 96.7% sensitivity and 83.3% specificity) than that of WLI (0.66 AUC, 66.7% sensitivity and 60.0% specificity) [94]. The research indicates that AI aiding different endoscopies to diagnose *H. pylori* infection can achieve acceptable accuracies in preclinical stage and more efforts in need to promote the real time endoscopic diagnosis directly in the future.

## 4. Noninvasive Tests

Conventional noninvasive tests commonly include UBT, *Hp* SAT and serological test.

UBT is the best recommended noninvasive test in the test-and-treat strategy and verifying test after eradication [28]. It can be divided into ^13^C-UBT and ^14^C UBT. The ^13^C-UBT is the best approach to diagnose *H. pylori* infection because of its simplicity, high accuracy, and being less affected by focal distribution of *H. pylori*. ^14^C UBT cannot be used in children and pregnant women due to the fear of its radiation [95,96,97]. Two meta-analyses showed excellent performances of ^13^C-UBT. One reported 95% sensitivity and 95% specificity, as well as ^14^C UBT with 95% sensitivity and 95% specificity [98], and another one reported 96% sensitivity and 94% specificity, as well as ^14^C UBT with 97% sensitivity and 91% specificity [99]. There are some restrictions of using UBT, including discontinuing PPI for at least 2 weeks and antibiotics and bismuth compounds for at least 4 weeks because of their anti-*H. pylori* activity and the decreasing load of *H. pylori* [100,101], and some possible specific conditions (peptic ulcer bleeding [102], gastric MALT lymphoma [24], and severe gastric atrophy and IM [103]), as well as detection value close to the cutoff value [104]. While H2 receptor antagonists without anti-*H. pylori* activity have minimal effect on the sensitivity of UBT and antacids do not impair the sensitivity of UBT or SAT [105].

*Hp* SAT is considered as an alternative in detecting *H. pylori* and retesting after eradication, as well as diagnosing *H. pylori* infection in children and postgastric surgery patients [106,107]. Two types of SAT include polyclonal antibodies based on enzyme immunoassay (EIA) and monoclonal antibody on immunochromatography (ICA) [108]. One meta-analysis reported 91% sensitivity and 93% specificity of SAT [109]. Another one showed 92.4% sensitivity and 91.9% specificity [110]. For the confirmation of *H. pylori* eradication more than 4 weeks after therapy, 86% sensitivity and 92% specificity was shown in one meta-analysis [109], and 88.3% sensitivity and 92% specificity in another one [110]. One study demonstrated 91.6% sensitivity and 98.4% specificity of the monoclonal SAT and 87.0% sensitivity and 97.5% specificity of the polyclonal SAT [111]. Monoclonal SAT was more accurate after eradication therapy than polyclonal SAT [109,112]. There are some restrictions when using Hp SAT. Low density of *H. pylori* in the stomach and a low antigen load in the stool are considered as the most common factors causing false negative tests which can be caused using bismuth, PPIs or antimicrobials, unformed or watery stool samples and the interval time after eradication [113,114]. Also, temperature and the interval between stool sample collection and measurement also affect the results of SAT [115].

Serological test can be used for children and some specific conditions (peptic ulcer bleeding, gastric MALT lymphoma, severe gastric atrophy, and IM and the use of PPI or antibiotics), which may lead to a low bacterial load associated with false-negative results. It is not recommended for the diagnosis of an active *H. pylori* infection and detecting *H. pylori* after eradication because the antibodies may remain positive for decades after *H. pylori* eradication [116,117]. Positive results cannot distinguish between active infection and past exposure to *H. pylori*, and further confirmation by other tests is required. The detection of specific *H. pylori* antibodies in urine and saliva has no current role in patient management but can be helpful for epidemiological studies [118]. Serological tests can be used only after validation, proposed in the Maastricht V/Florence Consensus Report [28], as the accuracy depends on the antigen used in commercial kit and the prevalence rate of specific *H. pylori* strains employed as the source of antigen in different geographic locations, as well as the cut-off values. A meta-analysis including 34 studies with 4242 participants (2477 had *H. pylori* infection). A threshold of >7 units/mL was used in two studies involving 97 participants with 98% sensitivity and 71% specificity, and two studies involving 234 participants used a threshold of ≥300 units with 91% sensitivity and 86% specificity [98]. In Kyoto global consensus report on *H. pylori* Gastritis, serological tests (pepsinogen I and II and *H. pylori* antibody) are useful for diagnosing chronic gastritis and gastric atrophy and identifying individuals at increased risk for gastric cancer [119].

## 5. Recent Advances of PCR: QPCR and DPCR

Polymerase chain reaction (PCR) presently contains three types: conventional PCR, quantitative real-time PCR (qPCR), and digital PCR (dPCR). PCR amplifies DNA and generates several millions of copies of a specific segment of DNA from a minute amount of starting material. qPCR is based on PCR and measures the amount of PCR product after each round of amplification using a fluorescent readout. dPCR enables the absolute quantification of target nucleic acids present in a sample and alleviates the shortcomings of qPCR [120]. PCR methods can detect *H. pylori* in different specimens such as human saliva, stool, gastric juice, and biopsies and dental plaques. Two kinds of housekeeping genes including 16S rRNA and 23S rRNA were applicated in clinical detection of *H. pylori* infection and antibiotic resistance. One meta-analysis demonstrated the DOR of genes, and their performance ranking used stool PCR test was as follows: 23S rRNA 152.5, 16S rRNA 67.9 and glmM 68.1 [121]. Several studies of qPCR testing gastric biopsy, gastric juice, or stool showed more than 90% or 95% sensitivity to detect *H. pylori* and 100% sensitivity to test antibiotic resistance in patients with dyspepsia and similar or higher diagnosis ability compared with that of histological methods [122,123,124]. For example, a prospective multicenter study including 1200 adult patients compared qPCR performed on stool samples to detect *H. pylori* glmM gene and mutations in 23S rRNA conferring clarithromycin resistance with culture/E test of two gastric biopsy samples. It demonstrated 96.3% sensitivity, 98.7% specificity, and 98.2% accuracy for detecting *H. pylori* by qPCR and 100% sensitivity, 98.4% specificity, and 98.7% accuracy for detecting resistance to clarithromycin [125]. On the aspect of dPCR, studies simultaneously quantified *H. pylori* clarithromycin-resistant and -susceptible 23S (A2142G, A2142C, and A2143G) and 16s rRNA gene alleles in gastric biopsy and stool samples using droplet digital PCR (ddPCR) and indicated that ddPCR could detect *H. pylori* and its clarithromycin resistance-associated genotypes and might aid in immediately testing H. pylori status after eradication [126,127,128]. ddPCR was also useful in detecting low-density “occult” *H. pylori* infection in a significant proportion (36%) of patients diagnosed as negative by conventional methods [129]. Nested dPCR, as another type of dPCR, was also useful to test clarithromycin resistance performed on stool samples in middle school students [130]. As qPCR and dPCR, especially the latter one, are characterized as high sensitivity and the ability to test antibiotic resistance in less time, clarithromycin resistance test by PCR is recommended in the management of *H. pylori* infection. It may be also useful to find false-negative results caused by low density of *H. pylori* or its antigen. There are also some issues that still need our attention, such as the target genes to design the primer sets, commercial kits to extract DNA from stool, and false-positive results.

## 6. Conclusions

*H. pylori* infection is chronic and prevalent. Early detection and eradication can abolish the aggregation of chronic inflammation and atrophy and prevent GC. There are some challenges in clinical practice such as false-negative results, increasing antibiotic resistance, decreasing eradication rate, and poor retesting rate. In this review, we concluded present diagnosis methods mainly based on invasive endoscopy and noninvasive laboratory testing. From the view of endoscopy, some advances such as ME, LCI, and AI demonstrated better diagnostic powers than that of WLI to reduce the false-negative results and contribute to the present targeted biopsy and the future achievement of direct endoscopic diagnosis without biopsy. Meanwhile, potential noninvasive methods real-time detecting *H. pylori* during endoscopy qualitatively and quantitatively to replace biopsy is also equally valuable to be studied and developed. From the view of laboratory testing, the development of PCR can be helpful in antibiotic resistance and eradication rate. Regarding retesting rate, it is necessary to simplify the routine (retesting at least 4 weeks after eradication), besides conducting followup and surveillance system. As PCR is characterized with a high sensitivity, it may be useful to detect *H. pylori* status immediately after eradication. Therefore, it is promising in better diagnosis and management of *H. pylori* infection and preventing gastric cancer with the development of technology optimization and innovation.

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
