# Peer review of "Diagnosis of Helicobacter pylori Infection and Recent Advances"

_diagnostics, 2021, doi:10.3390/diagnostics11081305_

Round 1
Reviewer 1 Report
The manuscript entitled "Diagnosis of Helicobacter pylori infection and recent advances" is well structured, the references are adequate and support the written work.
However, I have some suggestions for the authors:
1) The manuscript is a reviwe, therefore it would be advisable to add images;
2) Being a review talking about diagnosis, it is necessary to add a section on the most advanced methods that aim to also understand the epigenetic changes that the bacterium can cause (doi: 10.1097 / MD.0000000000020761; doi: 10.3390 / biom9060237; doi: 10.3748 /wjg.v25.i32.4629)
3) Finally, it is necessary to add a section of the treatments that are carried out post diagnosis today and of how patients under therapy are monitored (doidoi: 10.1097 / CM9.0000000000000618: 10.1371 / journal.pone.0222295, DOI: 10.1111 / eci.12857 , doi: 10.1080 / 14787210.2016.1178065, doi: 10.1007 / 5584_2019_367)
Author Response
REVIEWER 1
Comments to the Author
- The manuscript is a review; therefore it would be advisable to add images;
Thank you for the comment. There is a graphic abstract to illustrate the main content of the manuscript.
- Being a review talking about diagnosis, it is necessary to add a section on the most advanced methods that aim to also understand the epigenetic changes that the bacterium can cause (doi: 10.1097 / MD.0000000000020761; doi: 10.3390 / biom9060237; doi: 10.3748 /wjg.v25.i32.4629).
Thank you for the comment very much. we have made some supplement on epigenetic changes caused by H. pylori in the introduction part. However, at present these changes detection is still in the phase of basic research and related methods have not been applied in clinical practice, which need further research.
- Finally, it is necessary to add a section of the treatments that are carried out post diagnosis today and of how patients under therapy are monitored (doidoi: 10.1097 / CM9.0000000000000618: 10.1371 / journal.pone.0222295, DOI: 10.1111 / eci.12857 , doi: 10.1080 / 14787210.2016.1178065, doi: 10.1007 / 5584_2019_367).
Thank you so much for the comment. Present principle of eradication treatment has been added in the part “2. Challenges of H. pylori management and recommended detecting strategy”.

Reviewer 2 Report
The manuscript entitled „Diagnosis of Helicobacter pylori infection and recent advances” is interesting and presents the latest developments in the diagnosis of H. pylori infection both in terms of endoscopy and non-invasive methods. The paper fits very well with the profile of the journal. It summarizes in a condensed way the knowledge on diagnosis of H. pylori infections.
Minor comments:
- Please add italics style in the name of microorganism (Helicobacter pylori) in whole manuscript and in the references.
- Please add the explanations of abbreviations in the introduction and sections 3 and 5 of the manuscript, lines 25, 38, 145, 352.
Author Response
REVIEWER 2
Comments to the Author
1. Please add italics style in the name of microorganism (Helicobacter pylori) in whole manuscript and in the references.
Thank you for the kind and careful comment. All of them have been italicized.
2. Please add the explanations of abbreviations in the introduction and sections 3 and 5 of the manuscript, lines 25, 38, 145, 352.
Thank you for the kind and careful comment. The explanations of abbreviation have been added. Some explanations are added in the introduction part after combining with other comments.
